# Geodynamic Evolution and Metallogeny of Archaean Structural and Compositional Complexes in the Northwestern Russian Arctic

**Nikolay E. Kozlov** [1,*] , **Nikolay O. Sorokhtin** [2] **and Eugeny V. Martynov** [1]

1 Geological Institute of the Kola Science Centre RAS, 14 Fersman Street, Apatity 184209, Russia; mart@geoksc.apatity.ru
2 P.P. Shirshov Institute of Oceanology RAS, 36 Nakhimovsky Prospect, Moscow 117997, Russia; nsorokhtin@mail.ru
* Correspondence: kozlov@geoksc.apatity.ru; Tel.: +7-81555-79-656

**Abstract:** This paper highlights the geodynamic evolution of the early Precambrian rock associations in the northwestern part of the Russian Arctic where the rocks are exposed in the Kola region (northeastern Baltic Shield). The evolution is shown to predetermine the metallogenic potential of the area. It is emphasized that the Earth's evolution is a non-linear process. Thus, we cannot draw direct analogies with Phanerozoic time or purely apply the principle of actualism, which is still widely used by experts in Precambrian geology to study the premetamorphic history of ancient deposits. In both cases, the principles should be adjusted. This article provides a novel technique for reconstructing geodynamic regimes of protolith formation in the early Precambrian. The technique identifies changing trends in geodynamic regimes during the formation of the Archean structural and compositional complexes in the Kola region. These trends fit into the earlier suggested general scheme of their formation, thus enhancing its reliability. The metallogeny of the ore areas is specified. The results of the geodynamic reconstructions explain most of the location patterns of minerals within the Kola region. Thus, the authors consider the metallogenic forecast based on geodynamic reconstructions to be a promising trend for further research.

**Keywords:** evolution of the composition; basic rocks; Precambrian; search of trend differences; geodynamic evolution; metallogeny

## 1. Introduction

The basis for studying the geodynamic evolution of Early Precambrian rock assemblages in the northwestern Russian Arctic that crop out within the Kola region (northeastern Baltic Shield) is provided by data on their compositional variations with time. The authors used geological–structural and isotope–geochemical methods combined with geophysical investigations to outline stages of the most ancient structures in the region [1–5]. Notably, the results of geochronological analyses, however accurate, still provide ambiguous interpretations of ages. Thus, the age of the Patchemvarek metagabbro massif (2935 ± 6 Ma) is considered to be the beginning of the Kolmozero-Voronya Greenstone Belt formation [4], or the age of the "best-preserved fragments of the ancient protolith in pregranulitic and pregranitic associations of the Murmansk microcontinent" [2]. The time of the protolith formation in basic rocks of the Lapland (Lapland-Kolvitsa) Granulite Belt is also in doubt. Some authors [1,2] believe the protoliths to be Paleoproterozoic, while others [6–8] have argued that at least a few of the rocks in the belt section are Archean. Certain supporters of the first concept consider the protoliths to be Paleoproterozoic with the proviso that mafic granulates should not be older than the Paleoproterozoic gabbro-anorthosite massifs [2].

At the same time, it is clear that the volcanic evolution can only be explored by ranking the studied rock assemblages according to the time of the formation of their protoliths. Understanding that the future of this issue belongs to geochronology, we assign some certain role in this to the methods relied on applying the data on the rock composition that allow spotting trends in compositional changes in rocks and their associations over time, as shown in previous studies ([7–11] etc.). It is also important that these methods concurrently describe such trends using statistically evaluated indicators. Though there are many evolution models, data on the composition of metamorphites composing early Precambrian domains have rarely been considered. To close the gap, we have worked for a long time to address the above issue. In this respect, the first detailed survey [12] and one of the latest papers dedicated to this issue should be noted [13].

## 2. Geological Setting

A detailed description of the Archean geology of the northeastern Baltic Shield has been provided [10]. Structural and compositional complexes originated here in the early Mesoarchean era (probably in the Paleoarchean era) and formed by the end of the Mesoarchean era to the beginning of the Neoarchean era. This period marked the genesis of two major segments (lithospheric plates) of the continental crust, i.e., the Belomorian (Belomorsky mobile belt) and Kola granulite-gneiss areas, which show unique patterns of crustal evolution. The later stage covers the time span from the Mesoarchean and Neoarchean border up to date and indicates the time when the united Archean Karelian-Kola lithospheric plate existed. As a result, a united collision structure formed within the northeastern Baltic Shield through collided basic-ultrabasic domains and areas of the continental crust [10,14–16].

The Kola Granulite-Gneiss Area (GGnA) occupies the northeastern part of the Baltic Shield (Figure 1). It borders the Belomorskaya mobile area in the south and southwest and is overlapped by the Russian Plate cover in the east and southeast. In the north and northeast, the Kola GGnA is constrained by the Karpinsky fault in the Barents Sea. In general, it shows a typical mosaic structure of the continental crust and consists of structurally different domains (Figure 1). There are six in the region: The Murmansk, Kola-Norwegian, Keivy, Eastern Kola, Chapoma, Umba, and Tersky domains. Most of these formations are divided by narrow linear belts of the greenstone type, i.e., the Kolmozero-Voronya (Titovsko-Kolmozersky), Sergozero-Strelnensky, and Lapland granulite belts. There are a few points of contact where two closely jointed crustal domains are marked by specific structural and compositional parageneses [10,16].

At present, despite the extensive and detailed investigation of the Archean structural and compositional complexes of the Kola GGnA, reconstructing its earlier stages is still a challenge, and datings of its generation are obscure [8–11,13,17–21]. We can clearly define structural-metamorphic complexes that show the upper Archean collision of the continental crust, which is well-observed in the eastern part of the Baltic Shield. Recent detailed research of the most ancient complexes in the Kola-Norwegian domain yield no evidence of relic parageneses at the earliest stages of the continental crust generation.

Generally, the Kola-Norwegian domain has an imbricated thrust structure that formed in the late Archean era and at the Rebolian folding phase. However, dome-shaped folds of the same age still occur in the southwestern part of the domain, which indicates that intensive granitization prevails in this area (the Priimandrovsky district). The Archean complexes are represented by repeatedly and heterogeneously metamorphosed formations from amphibolite facies of the high-temperature stage to low-gradient granulite facies.

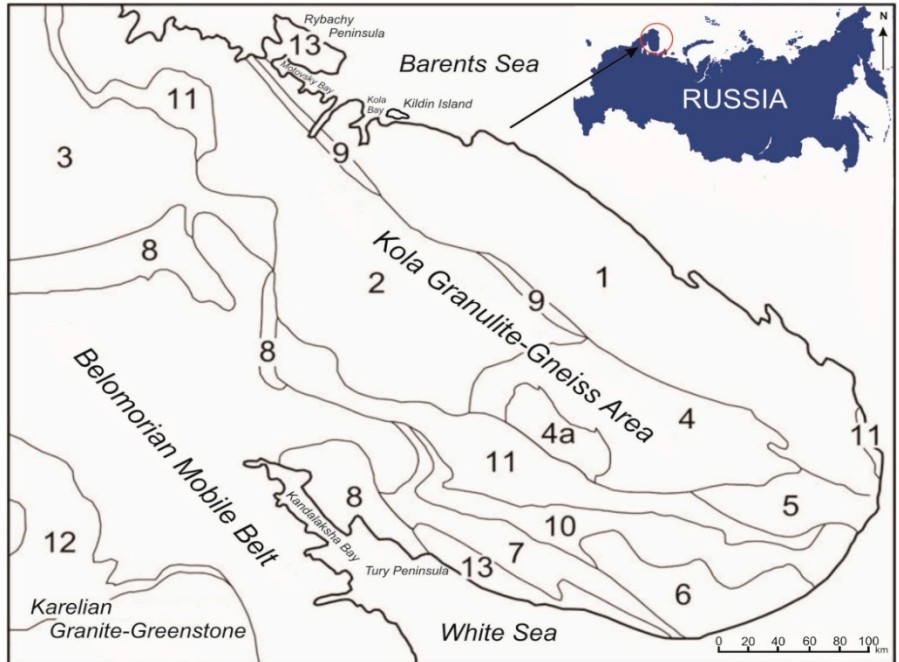

**Figure 1.** Zoning of the early Precambrian geostructural elements of the Earth's crust in the northeastern Baltic Shield (Modified from [10]). Crustal domains: 1—Murmansk, 2—Kola-Norwegian, 3—Lotta, 4—Keyvy, 4a—Verkhneponoysky, 5—Eastern Kola, 6—Chapoma, 7—Tersky. Greenstone and granulite belts (8–10—Archaean; 11, 12—Neoproterozoic): 8—Lapland-Kolvitsa; 9—Titovka-Kolmozero (Kolmozero-Voronya); 10—Sergozersky–Strelninsky; 11—Pechenga-Imandra-Varzuga-Ustponoysky; 12—Northern Karelian; 13—Riphean rift and continental marginal sediments.

The Belomorian Mobile Belt (MB) is a separate crustal block that occurs between two major tectonic structures of the Baltic Shield, i.e., the Karelian granite-greenstone area (GGrA) and the Kola granulite-gneiss area (GGnA) (Figure 1) [10]. The width of the junction zone with the latter varies from 2–3 to 15–20 km and is well-detected according to geophysical data. In the magnetic field, the border is marked by chains of linear suture-oriented anomalies. The suture zone has linear-type structures. Here, margins of the Kola block thrust over the Belomorian block. Most of the contact is marked by the Lapland-Kolvitsa Granulite Belt. It comprises basic metamorphic rocks altered both in amphibolite and in granulite facies, including the eclogite subfacies of granulite facies [8,18,22].

## 3. Materials and Methods

The order in which the Archean complexes (rock assemblages) of the northeastern Kola region formed has been previously outlined during the analysis of the composition of the metabasites [13]. Since the results of these studies provide the basis for further reconstructions, we deem it necessary to briefly point out the applied methods and present the obtained results.

The behavior of the chemical rock compositions was described using the method of modeling the trend of variations in chemical compositions of rocks that compose the studied rock assemblages as compared to the partial order entered by the researcher towards the combination of these assemblages. The algorithm applied to search for this trend has been mainly highlighted in Russian scientific journals and will be briefly described here. It shall also be noted that all the algorithms applied in this research are published in international scientific sources. Suppose $Z = \{Z_i\}$ represents a set of n-dimensional random variables, and partial order '$\rightarrow$' is given on set $Z*Z$. If c is an n-dimensional vector of unit length, the scalar product $(c, Z_i)$ is a one-dimensional random variable. This random variable may be described by its mathematical expectation, $M(c, Z_i)$. Each of these random variables $(Z_iZ)$ are represented by a sample of n-dimensional vectors $(V_i = \{v_{ij}\})$ in Euclidian space, $R^n$ (i.e., the sample $V_i$ describes the chemical compositions of rocks composing the corresponding rock

assemblage). To compare the mathematical expectations of the random variables (c, $Z_i$ and c, $Z_j$), the Puri–Sen–Tamura rank-order test for the equality of means is proposed to be used [23,24]. The choice of the specified statistical criterion is defined by both its stability against the violation of the normality (or even unimodality) condition for the random variable distribution and against the availability of anomalous observations in the samples.

For the above criterion to be used, it is necessary to estimate the means (median Me(c, $Z_i$) is chosen as the estimator) and calculate the Puri-Sen-Tamura statistics $\Lambda((c, Z_i),(c, Z_j))$ for all pairs of random variables ($Z_i$, $Z_j$). The statistical modeling (where the set of variables is specified by ratio '→') involves the search for the n-dimensional vector c of unit length, for which, at a chosen significance level, $\delta$, the following conditions are satisfied Equations (1) and (2):

$$Me(c, Z_i) < Me(\mathbf{c}, Z_j) \tag{1}$$

$$\Lambda((c, Z_i),(\mathbf{c}, Z_j)) > \chi^2(\delta) \tag{2}$$

where $\chi^2(\delta)$ denotes a value of $\chi^2$-inverse distribution for the significance level $\delta$ of all pairs $<Z_i,Z_j>$ so that $Z_i{\rightarrow}Z_j$.

The trend modeling task lies in the approximation of the given partial order by linear function P: Z→R, where R is a set of real numbers. The approximation quality is estimated by the value of the function: $J(P) = \max_c \min_{(i,j)} \Lambda(\{(c,v_{ik})\}_k,\{(c,v_{jl})\}_l)$, where on the set of pairs $\{(Z_i, Z_j)\}$ so that $Z_i{\rightarrow}Z_j$.

Thus, the modeling of the trend in differences of chemical compositions of rocks that compose the studied rock assemblages as compared to the partial order entered by the researcher towards the combinations of these assemblages reduced itself to a classical functional J(P) optimization task on the given set in view of the entered constraints. The optimal solution was searched for using the known Nelder–Mead algorithm [25].

Vector c, which is further referred to as the partial-order factor, describes the common trend in the behavior of the chemical compositions in relation to the given partial order.

As an illustration of the above, we shall provide the following hypothetic example (Figure 2). In this case, the coordinate axes convey no specific geological meaning. It is only valuable here that the space is two-dimensional, and thus, it is enough to preset two arbitrary axes for introducing a coordinate system in this space. Suppose that some combination of rock assemblages {A, B, C, S, H, D, E, K, M} is given for the two-dimensional feature space. Each of the assemblages is represented by a figurative point (for example, a means point). The partial order (for example, 'younger' or '→') in this combination is displayed as a graph, namely B is younger than S, C is younger than D, D is younger than E and H is younger than K. It is required to find the linear function P implemented as axis F specified by vector c in the feature space so that functional J(P) takes on a minimum value in the ratio (B→S, C→D, D→E, H→K).

To find the partial order that describes the compositional changes in the Archean metabasites from younger to older formations, the following conditions were formulated based on the geological setting and geochronological data [2,8,10,21]:

1.  The Titovka-Kolmozero (Kolmozero-Voronya) Belt is the youngest when compared to the Murmansk and Kola-Norwegian domains, since its rock assemblages formed at the pre-metamorphic stage as a result of interaction of the domains at their border;
2.  The Lapland-Kolvitsa Granulite Belt, regardless of its age, is younger than the Lotta domain and Belomorian Mobile Belt, due to the interaction of which it was emplaced as a volcanogenic-sedimentary complex;
3.  According to available data, rocks of ancient complexes in Karelia, Canada, and Greenland formed at an earlier stage of the Earth's evolution as compared to the Archean metamorphites of the Kola region.

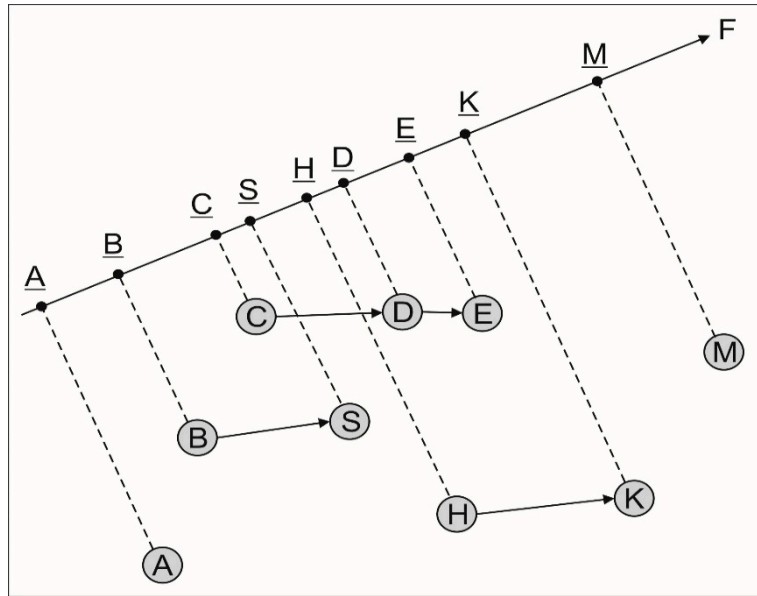

**Figure 2.** Partial order relation presented as a graph: B precedes ($\rightarrow$) S; C$\rightarrow$D; D$\rightarrow$E; H$\rightarrow$K. Location of A and M, as well as B and S regarding C, D and E, as well as H and K on axis F is not introduced initially [13]. Axis F describes the location of rock assemblages towards to each other with time.

It shall be noted that the ambiguous interpretation of the protolith age of the Lapland-Kolvitsa granulite belt, which was mentioned earlier in the introduction, does not affect the second condition since it is satisfied when accepting both the Archean and Paleoproterozoic time of emplacement for this structure.

The formulated conditions define the following partial order:

1. The Murmansk Domain precedes ($\rightarrow$) the Titovka-Kolmozero Belt, and the Kola-Norwegian Domain precedes ($\rightarrow$) the Titovka-Kolmozero Belt;
2. The Lotta Domain precedes ($\rightarrow$) the Lapland-Kolvitsa granulite belt, and the Belomorian Mobile Belt precedes ($\rightarrow$) the Lapland-Kolvitsa granulite belt;
3. Each of the oldest rock assemblages in Karelia, Canada, and Greenland precede ($\rightarrow$) each of the Archean rock assemblages in the Kola region.

As a result, we found the partial order factor ($F_1$) which describes the trend in the compositional changes of the metabasites in the studied rock complexes. Alongside, the above formulated conditions have to be concurrently fulfilled with the differences in rocks for each task to be significant at the 5% significance level (Figure 3). The metamorphites of the Archean complexes are clearly seen to group on it and form sets that we code-named groups A, B and C.

Notably, searching for factor $F_1$ was based on data on the composition of rock assemblages where relative age can be defined quite accurately based on geological data. We did not introduce the locations of complexes listed in different conditions of the task as well as other structures on the factor. As far as we remember, the location of most Archean complexes on factor $F_1$ does not contradict the sequence of formation during the northeastern Baltic Shield evolution. Its pattern has been suggested by a number of authors on the basis of complex rock study compared to geophysical and geochronological data [2,8,20].

Since any petrogeochemical reconstructions are probabilistic, the only way to increase their reliability is to introduce new data that would enlarge the study area. As long as these reconstructions do not clash with the already obtained results, we seem to take a step that approaches high probability, but never reaches it. In this research, such a step was taken in studying the evolution of the geodynamic regimes of metavolcanite formation in the Archean era in the Kola region.

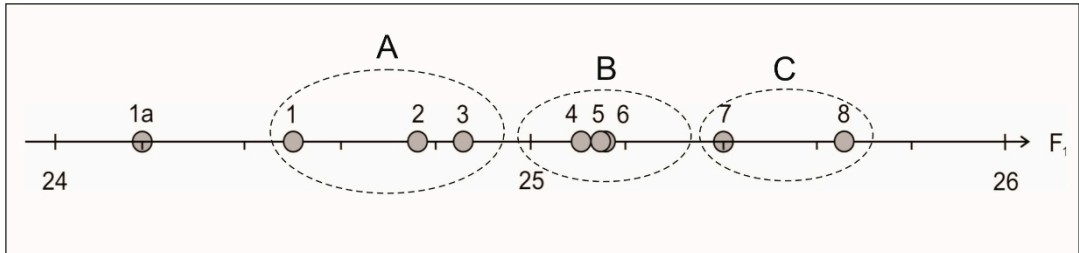

**Figure 3.** Location of compositions of Archaean metabasites in the Kola region compared to formations in Karelia, Canada, Greenland, on factor $F_1 = 0.34SiO_2 - 0.46TiO_2 + 0.25Al_2O_3 + 0.19\Sigma Fe + 0.19MnO - 0.07MgO + 0.38CaO - 0.24Na_2O - 0.57K_2O$ [13]: 1a—ancient formations in Karelia, Canada, Greenland, 1–8—domains of the Kola region: 1—Keyvy, 2—Lotta (Allarechka and Notozero complexes), 3—Chapoma, Tersky and partly Umba, 4—Kola-Norwegian, 5—Murmansk, 6—Belomorian Mobile Belt; 7–8—Archaean belts of the Kola region: 7—Lapland-Kolvitsa, 8—Titovka-Kolmozero. A, B, C—enlarged groups to be described further.

In this study, data on the geology and mineral composition of the early Precambrian complexes were collated and compared. As with the previous task, basic rocks were studied to make various reconstructions. Protoliths of their metamorphosed analogues are best recognized by their original features. Furthermore, they are widespread within most of the early Precambrian structures of the region.

Previously, we defined regular and statistically significant differences in the petrochemical characteristics of the Precambrian metabasalts and Phanerozoic basalts (Figure 4), which provide evidence that processes of the Earth's evolution were non-linear [9]. The nonlinearity of the evolutional processes is attested by the following fact established in this research. As a result of studying the mutual arrangement of reference images (Phanerozoic basalts) and rock assemblages of Precambrian metabasalts in the feature space, it turned out that the shifting of their petrochemical features could not be statistically reliably described using one linear function. This meant that at least two linear functions needed to be applied to describe such a shift, as shown in Figure 4.

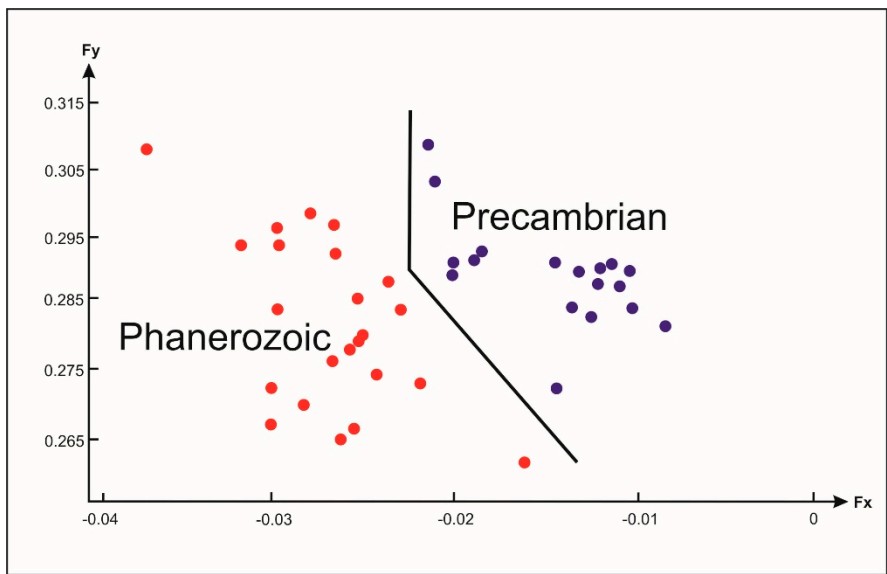

**Figure 4.** Location of median values of compositions of basic rocks from different Phanerozoic (red dots) and Precambrian (blue dots) rock assemblages [9] on diagram $Fx - Fy$, where: $Fx = -0.1SiO_2 - 0.82TiO_2 - 0.11Al_2O_3 + 0.34FeO + 0.06MgO + 0.03CaO - 0.28Na_2O + 0.32K_2O$; $Fy = 0.45SiO_2 - 0.28TiO_2 + 0.21Al_2O_3 + 0.22FeO - 0.06MgO + 0.24CaO - 0.27Na_2O - 0.75K_2O$.

In addition, the conventionally applied methods for the reconstruction of geodynamic regimes are often based on reference Phanerozoic rock assemblages that are represented by statistically invaluable samples. This particularly concerns reconstruction methods that apply various diagrams [26–36]. This is the reason why we were quite careful when using such methods for modeling geodynamic regimes. Thus, the principle of direct analogies with the Phanerozoic cannot be applied for the reconstruction of the Precambrian geodynamic settings without making necessary adjustments, i.e., considering the shift of rock images in the feature space when moving from the Precambrian to the Phanerozoic time and the representativity of the applied reference Phanerozoic rock assemblages.

In this study, we compared Precambrian and Phanerozoic rock complexes that were genetically linked to certain regimes on the basis of the idea of the specific nature of the Precambrian period in the Earth's evolution, and on some commonness in the geodynamic settings manifested throughout the whole geological history, i.e., homolog rows of geodynamic regimes [7]. This comparison was performed on more than 1100 bulk rock chemical analyses of Precambrian metamorphites and about 1100 bulk rock chemical analyses of Phanerozoic magmatic rocks.

We analyzed data on the abundance of certain varieties of original (premetamorphic) rock associations in the study areas. This work was based on reconstructions of the original features of rocks according to the petrogeochemical characteristics of compositions. For this, the method described in Reference [37] was applied. Like other techniques elaborated for these purposes, this method provides no solution for this task (for felsic metamorphites, first of all). This was solved in two ways. When points of rock composition occurred within the field covering sedimentary and magmatic formations on the reconstruction diagrams, such rocks were all referred to the first group in one comparison variant. In another, they were referred to the second group. Next, patterns of reconstruction variants were compared. We assumed all intermediate decisions to be within these extreme decisions. Once patterns of both variants coincided, the final conclusion on a spreading character was made.

As stated above, systematic differences in the composition of the Precambrian rocks and their Phanerozoic homologs constrain the application of numerous diagram methods to reconstruct early Precambrian geodynamic regimes. The authors have worked on this issue for a long time [7,9]. To obtain more accurate results, we elaborated on a method that describes the mentioned non-linearity even more efficiently than previously used techniques. Instead of 10 petrogenic elements ($SiO_2$, $TiO_2$, $Al_2O_3$, FeO, $Fe_2O_3$, MnO, MgO, CaO, $Na_2O$, and $K_2O$), we used eight parameters of the chemical composition, turning the FeO, $Fe_2O_3$, an MnO parameters into a new one, provisionally named $\sum$FeO. We described the shifting of the Precambrian rock assemblages regarding the Phanerozoic standards in 8D feature space (the Precambrian rock assemblages and Phanerozoic standards shown by multiple figurative points) in terms of a quadric surface dividing them. The criterion to determine this surface as dividing was the statistical significance of difference between multiple figurative points of each rock assemblage with a multitude of projections of these points on the surface. Multiple figurative points of the Precambrian and Phanerozoic rock assemblages were arranged on either side of this surface (Figure 5). Furthermore, we used an optimal dividing surface for which the minimal proximity of the whole rock assemblage totality to the surface was maximal. Here, we applied a quadratic decision rule as follows Equation (3):

$$(z - Y)S_2^{-1}(z - Y)' - (z - X)S_1^{-1}(z - X)' \geq 2ln\frac{|S_1|^{1/2}}{|S_2|^{1/2}} \tag{3}$$

where $S_1$ and $S_2$ are the estimates of covariance matrixes, and $X$ and $Y$ are averages of the Precambrian and Phanerozoic rock assemblages.

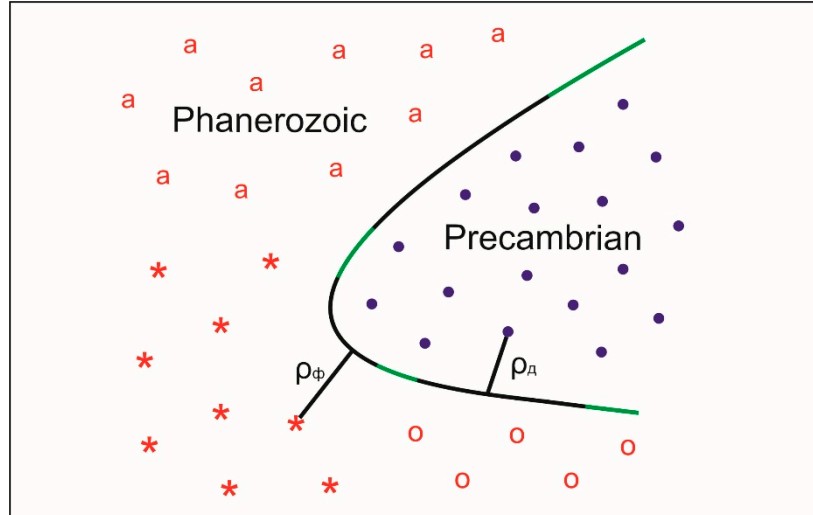

**Figure 5.** Schematic reconstruction of settings based on a quadric surface: **a**, **\***, **o** for the Phanerozoic rock assemblages formed in different geodynamic settings, • for the Precambrian rock assemblages; $\rho_\phi$ for the proximity of a certain Phanerozoic rock assemblage represented by a set of figurative points in a space of chemical rock composition which has parameters similar to the simulated quadratic surface in relation to chosen metrics, and $\rho_A$ for the proximity of a certain Precambrian rock assemblage to the surface respectively.

To determine the geodynamic settings of the studied Precambrian rock assemblage, we used multitude figurative points for the Precambrian rock assemblages and Phanerozoic standards on the designed dividing surface. Thus, we used different proximity degrees for the projection points of figurative points that indicated the chemical compositions of the Precambrian and Phanerozoic rock assemblages. It also allowed us to verify the obtained results with statistical criteria.

Since the distribution of figurative points of the rock assemblages does not comply with any of the known classic criteria, we applied nonparametric ones. In this case, we used the Puri-Sen-Tamura criterion [23,24] that is stable to a disrupted normal (and even unimodal) distribution of studied random variables and to anomalous observations in samplings. As noted, there are a number of proximity degrees. Choosing one of them is a challenge since there are no science-based arguments in one's favor. Therefore, the authors applied the following technique. First, the regime of the studied Precambrian rock assemblage origin was reconstructed using different proximity degrees. Next, the obtained reconstructions were compared. Once decisions for most of the applied proximity degrees coincided, the decision was considered accurate.

The study of many Precambrian rock assemblages showed that ancient complexes that formed in different geodynamic settings differed less than their Phanerozoic homologs. The reason for this is still unclear. However, we may state that when Precambrian complexes are geodynamically reconstructed, it is more accurate to consider changing trends in regimes than their complete analogues.

## 4. Results

Table 1 and Figure 6 provide reconstructed geodynamic regimes of protolith formation in Precambrian complexes of the Kola region. Since sampling in the Chapoma, Tersky and partly in the Umba domains was not representative enough (group 3 in Figure 3), it was not considered in the current research. It is impossible to refer geodynamic settings of protolith formation in any complex to a certain type because of the reasons provided in the previous section. As shown by a previous study (Figure 3), there is a clear pattern of interchanging regimes in geological evolution. Petrogeochemical characteristics of these regimes make them similar to traps that are closer to continental reefs. At the final stage of the formation of the Archean complexes, they become similar to young arcs. In the geological time span, such an interchange of regimes is possible within a small area.

Thus, a geodynamic evolution with varying plume and subduction magmatism ratios in one region was described for younger complexes in the central and eastern Arctic [38].

**Table 1.** Reconstructed formation settings of some Precambrian complexes.

| Groups, Figure 3 | NoNo, Figure 3 | Structures | TRAP | CR | MA | YA | DA | MOR |
|---|---|---|---|---|---|---|---|---|
| A | 1 | Keyvy Domain | **3.611** | 4.021 | 4.022 | 4.075 | 4.174 | 5.014 |
| | 2 | Lotta Domain | **2.667** * | 3.198 | 3.430 | 3.042 | 3.864 | 4.002 |
| B | 4 | Kola-Norwegian Domain | 2.448 | **2.278** | 3.265 | 2.308 | 2.791 | 2.781 |
| | 5 | Murmansk Domain | 2.549 | **2.361** | 3.145 | 2.371 | 3.074 | 3.144 |
| | 6 | Belomorian Mobile Belt | 2.549 | **2.270** | 3.050 | 2.349 | 2.833 | 2.775 |
| C | 7 | Lapland-Kolvitsa Belt | 2.535 | 2.135 | 2.982 | **2.117** | 2.675 | 2.435 |
| | 8 | Titovsko-Kolmozersky Belt | 3.555 | 3.284 | 4.045 | **3.170** | 3.855 | 3.275 |

* "distance" of Precambrian samplings to the respective standard (Phanerozoic) groups provisionally called "a proximity coefficient". The lower values of the provided coefficients (bold type), the closer compared rock assemblages are. Underlined figures indicate values of a minimal difference, if they insignificantly differ from the next-largest value with the significance level of $\alpha = 0.05$. Bold indicated values of a minimal difference. TRAP—traps, CR—continental rifts, MA, YA, DA—mature, young and developed arcs, respectively, MOR—mid-ocean ridges.

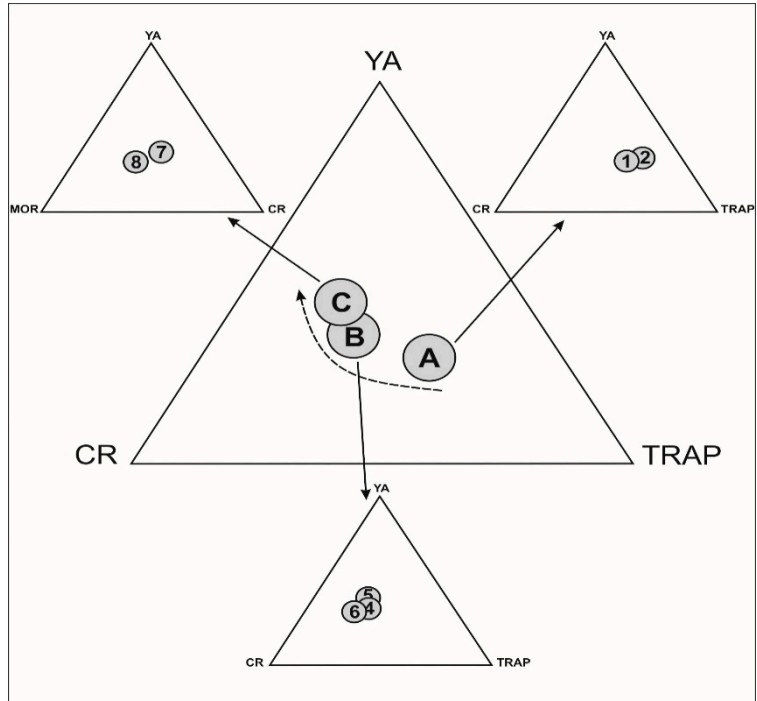

**Figure 6.** Location of composition points of the Archaean complexes in the Kola region on diagrams showing relative proximity to certain geodynamic regimes. Notations are specified in Table 1.

Many researchers believe that metabasites of ancient structures are close to rocks of trap formations that are genetically confined to plumes. Metabasites of younger structures are considered similar to the regimes typical of plate-tectonic processes [39]. This concept is rather interesting and matches the well-known concept that plate-tectonic processes substituted the plume-tectonic ones at early stages of the Earth's evolution.

Therefore, we may assume that disseminated rifting developed in the ancient crust during the early stages of formation of supracrustal complexes in the Kola region. The crust has features of the continental one. Relics of it have been possibly found in zircon cores in recent studies [40–42]. Thus, we suggest a more complicated multistage genesis of the continental crust within the Keivy domain when compared to other domains in the region. This was previously supposed in Reference [10] on

the basis of regional geological-geophysical data [10]. The Lotta domain metabasites are also the most similar to trap formations.

These data should be considered jointly with similar and different features in the mineralogical composition of rock associations of Archean domains in the Kola region. We made a tentative conclusion that its central part has a line composed of metamorphites of the Kola–Norwegian and Keivy domains. This line clearly differs from rocks of the Murmansk and Belomorian domains and has features of the Keivy domain [10]. This conclusion was verified based on a set of fact data obtained in the recent decade. More than 2100 bulk rock chemical analyses of metamorphic rocks of different compositions were made to verify this. The sampling range of metamorphites from the Chapoma, Tersky and part of the Umba domains was considerably extended when felsic to intermediate rocks were included as study areas.

In this study, the Keivy, Lotta, Chapoma and Tersky (also part of the Umba) domains were defined as similar (Figure 7). The Keivy domain is still the closest to the Kola–Norwegian domain. Notably, substances of the Lotta domain were more similar to those of the Chapoma, Tersky and part of the Umba domains. The similar composition of metabasites in these areas was noted earlier in Reference [10], when the evolution of Archean formations was studied. Thus, it proves the suggestion set by Ivanov et al. on the possible allocation of supracrustal formations of the mentioned domains as a single zone [17].

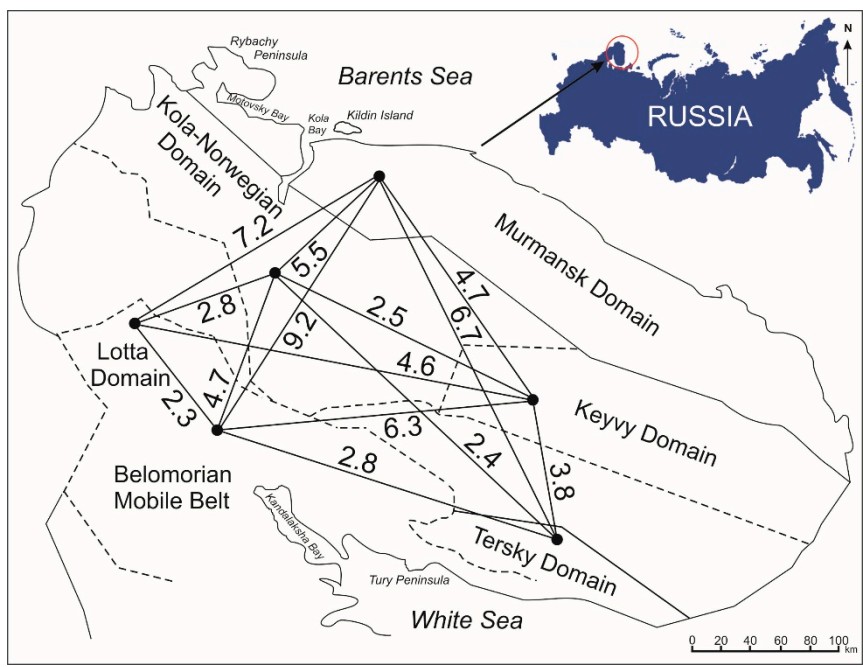

**Figure 7.** Compared mineralogical compositions of different Archaean structures in the Kola region. Structures are numbered as in Figure 3. Numbers near lines indicate a proximity coefficient of respective rock assemblages. The lower the coefficient of difference, the more the compared rock assemblages are similar.

Notably, the mineral composition of the Murmansk Domain metamorphites was significantly different from other structures in the region. In general, ratios of conditional proximity values indicating compositions of rock associations in different domains (Figure 7) were close to the data we obtained earlier ([10], Figure 3), but specified them greatly. Previously, we linked a minimal value of this coefficient to the fact that rock associations of the Murmansk domain belonged to the so called granite–greenstone areas (GGrA). Their ancient age of origin was also considered. As for rock associations from other domains, we remembered them belonging to the intermediate-type or granulite–gneiss areas (GGnA) [21].

With the new data, we may presume the coefficients of difference obtained for the studied complexes, just like previously described estimates of proximity to rock associations in Canada, Greenland and Northern Karelia, to reflect not so much the time of the protolith formation, as the geodynamics of processes within these structures, from the first stages of the protovolcanism and protosedimentation to further transformations at different levels of the Archean crust, which was suggested by Sorokhtin ([10], Chapter 5). The fact that there is a certain zoning within the Baltic Shield, i.e., GGrA rim GGnA and intermediate-type areas in the southwest and northeast, means this area deserves further research.

The ratio consistently changes within the studied domains, which are arranged according to the suggested chronological order of the formation of their protoliths, varieties, reconstructed metavolcanic (metavolcanites and metatuffites) and metasedimentary rocks (Figure 8). We believe this ratio to reflect the intensity of volcanism at certain stages of the region development. The Archean complexes of the northeastern Baltic Shield provide evidence of pulsating volcanic activity, as mentioned earlier [12].

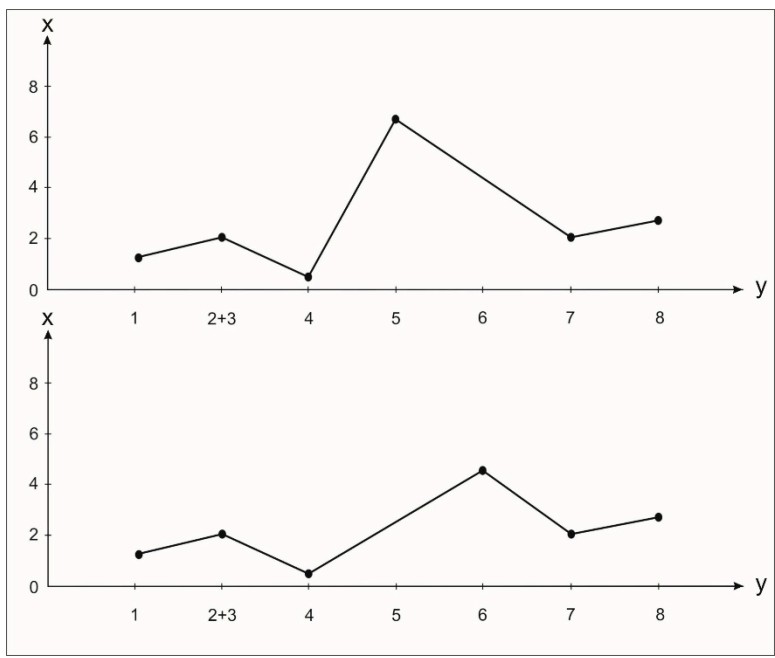

**Figure 8.** Varying ratios of metavolcanites and metasedimentary rocks distribution (axis x) in the studied complexes (axis y). Groups are numbered as in Figure 3. Constructed in two graphs, for the Murmansk Domain (group 5) and the Belomorian Mobile Belt (group 6).

## 5. Discussion

In the late Archean era, the geodynamic evolution of the northeastern Baltic Shield was associated with highly intensive structural metamorphism of the continental-crustal matter. Some new matter was added and border structures (collision sutures) were formed. These were marked by greenstone and granulite belts. As a result, a single collision complex occurred and united certain domains of the continental crust. These domains are rimmed by greenstone belts where the protooceanic lithosphere was taken in. Each of the crustal domains composing the collision zone had its own evolution that was dampened by further imposed events.

The obtained reconstructions are considered promising for the metallogenic assessment of the studied areas. As noted, the Paleoproterozoic PGE mineralization of the Kola region is likely to be linked with its Archean history [19,21]. When geodynamic reconstructions are compared to map data on deposits and ore occurrences in the Kola region [5], trends in the distribution of minerals and their association with the most ancient complexes in the region can be traced (Figure 9).

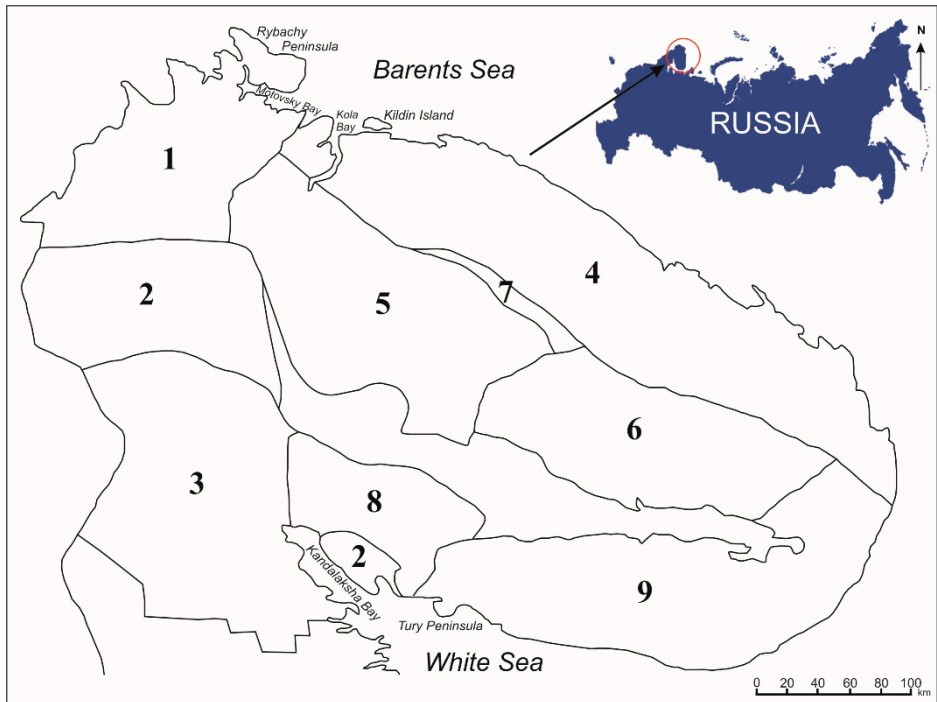

**Figure 9.** Metallogenic areas in the Murmansk region (Modified from [5]): 1—Pechenga-Allarechka area (nickel, copper, cobalt, sulphur, gold, silver, platinum, palladium, rhodium, ruthenium, iridium, selenium, tellurium, mica and ceramic pegmatites); 2—Lapland ore area (copper, nickel, vanadium, titanium, iron, manganese, molybdenum, graphite and gold, chromium, construction and decorative materials); 3—Belomorian ore area (deposits of muscovite and ceramic raw materials, sulfide (Cu-Ni) and Pt-bearing mineralization, raw materials for the aluminium industry); 4—Murmansk ore area (construction and decorative granite, ferruginous quartzites and ore occurrences of the thorium-uranium and uranium type); 5—Olenegorsk ore area (major deposits of ferruginous quartzites, apatite-silicate ores, apatite-magnetite ores of the Kovdor type, apatite-carbonate ores, tantalum-niobium ores, apatite and perovskite-titanium-magnetite ores, construction materials); 6—Keyvy ore area (ore occurrences of niobium, tantalum, zirconium, yttrium, thorium, uranium, tin, wolfram, vanadium and cobalt, major deposits of kyanite, abrasive garnet and amazonite, quartzites, high-grade vein quartz, muscovite, sillimanite, etc.); 7—Kolmozero-Voronya ore area (deposits and occurrences of gold, silver, molybdenum, lead, zinc, copper, nickel, iron, lithium, berillum, tantalum, niobium, caesium); 8—Kandalaksha ore area (titanium-magnetite and perovskite-titanium-magnetite ores with rare earths, phosphate raw materials, copper mineralization); 9—Tersky ore area (major occurrences of muscovite and ceramic pegmatites, decorative red sandstones, occurrences of molybdenite, carbonatites with apatite and rare metal mineralization, barite, amethyst, gold occurrences, diamond finds).

Previously, we linked increased concentrations of molybdenum and gold with the early Precambrian proto-island arc formations of the Lapland (Lapland-Kolvitsa) Granulite Belt [20]. Analyses of the metallogenic specification of the collision zones [43] suggest increased concentrations of titanium, manganese, vanadium, chromium and probably copper, nickel, and iron, which relate to the early Precambrian geodynamics of the formation of protoliths within the belt.

The authors recognize that increased concentrations of these elements in, e.g., the Lapland-Kolvitsa Belt stem from its geological history including its multistage metamorphism. Dehydration and anatexis of the protooceanic crust in collision zones are likely to follow a complicated multistage pattern. Spatial-temporal changes in metamorphic transformations mean that rock associations of the subducting protooceanic lithospheric plate were subjected to progressive metamorphism in the contact zone with the thrusting continent. They consistently passed all

stages of transformation, from the lowest to the highest ones. A mineralized gas-saturated fluid formed in these conditions and was transported up the faults. When the fluid cooled, it provided retrograde contact-metasomatic changes in surrounding rocks. Numerous ultrabasic protrusions and protoophiolites of the Lapland-Kolvitsa Granulite Belt were also subjected to retrograde metamorphism as they passed the peak of changes. All chemical reactions in the collision zones and the plate underthrust zones were irreversible and happened with the heat uptake or release in certain oxidizing-reducing conditions. The geological time played an important role in the above processes, balancing all physical-chemical parameters of the fold system.

Thus, increased concentrations of some elements can be related to the early geodynamic history of the Lapland-Kolvitsa Belt. The authors consider it to be the basic hypothesis of possible ore sources. This should be applied on a case-by-case basis, considering all rock transformations in certain ore occurrences. Similarly, increased concentrations of gold, silver, molybdenum, lead, zinc, beryllium, tantalum, niobium and possibly copper, nickel and iron in the Titovka-Kolmozero (Kolmozero-Voronya) suture zone can be linked to the geodynamics of its formation.

Iron ores could occur in the Kola-Norwegian domain since it is geodynamically close to continental rift areas that typically contain this element. In the southwestern part of the domain, there is a thick ferruginous volcanic-sedimentary complex stretching northwestwards along its margin. Though the complex was intensively and repeatedly subjected to later structural metamorphism, a trough-like mode of its occurrence can be detected. It contains ferruginous quartzites, metapelites, bipyroxene crystal schists, and amphibolites. Carbonate-bearing rocks are rare. Notably, bottom parts of the sections are mainly composed of amphibolites and are superimposed by biotite gneisses. Aluminous gneisses and ferruginous quartzites are typical for the top parts of the sections. We believe this sedimentation pattern to be evidence of the cratonization of the Earth's crust.

The fact that thick volcanic–sedimentary sequences of an iron-ore accretion prism formed at the edge of the Kola-Norwegian domain in the Neoarchean era requires study of the preconditions and patterns of ferruginous sedimentation at the Earth's surface. According to recent works, there are three main stages of iron ore sedimentation in the Earth's history [44]. The first stage is the most ancient and occurred 3.8–3.6 Ga ago. The second stage was 2.9–2.6 Ga ago and the third one was 2.3–1.7 Ga ago. The studied complex refers to the second stage of the iron ore sedimentation of mantle silicates. In the Proterozoic and Phanerozoic, it was subjected to the barodiffusive separation of the mantle material. The switch in mechanisms at the turn of the Archean and Proterozoic made the Archean geology basically different from that of other epochs and also predetermined the specific distribution of iron in the convective Earth's mantle.

We suggest that the Lotta domain copper–nickel occurrences within the Allarechka area formed partly because the domain is similar to trap formations with the same metallogeny. Both of these have similar geodynamic features of protolith formation. As for the Keivy domain, basic rocks that are similar to magmatic trap formations are minor and are likely to have no significant impact on the metallogeny of the area. Further processing of supracrustal formations greatly affected the domain metallogeny when weathering crusts formed in a steady period [12], followed by the regional metasomatism [45].

## 6. Conclusions

The obtained data were based on both new and previous research and prove the suggestion [10,11] that the oldest core of the Kola protocontinent formed in the northeast of the region and accreted further west and southwestwards. These data suggest the following development model of rock association in the Kola region.

There were two main stages of the continental crust evolution in the southeastern Baltic Shield. The first stage marked the origin of the continental crust of the Kola GGnA and Karelian GGrA. According to our data, the Kola GGnA originated when its core formed and is composed of basic rocks of the Keivy domain bottom, where the material was preserved as basic rocks of the Patchervtundra

and Lebyazhinskaya suits. The core was accreted in the southern and southwestern direction by formations that constituted the belt (intracratonic activation zone). This belt comprises supracrustal rocks of the Lotta, Chapoma, Tersky and part of the Umba domains.

In the west, complexes of the Kola-Norwegian domain developed at the next stage. Supracrustal complexes of the Belomorian domain formed in the further converging of the Kola GGrA and Karelian GGnA until they produced the single Archean Karelian-Kola lithospheric plate. Original volcanogenic formations of the Murmansk Domain could not occur at the first stage, but at the same time as the rock associations in the Belomorian domain. Since the Murmansk domain rocks have a peculiar composition, it is possible that before the Rebolian, the domain developed separately from the rest of the area and only joined other domains at its early stages.

Geological-structural and geophysical data [10] suggest that the Keivy microcontinent sunk to some depth by surrounding continental massifs that overthrusted it at a certain stage of its development. Orogenic belts formed around this structure later in the Neoarchean era at the Rebolian phase of folding. It complies with the idea [9,10] that since the Archean rock associations in the southwestern part of the region had been gradually accreting the older (Mesoarchean, probably Paleoproterozoic) core located in the north to northeast. At that time, the crust of the eastern Kola region was thicker and older and determined the occurrence of deeper magmatic sources here. In this, the rock associations of the Kola–Norwegian domain are likely to be the main removal area for the Keivy structure sediments, which made compositions of their rock associations similar. Thus, in the Neoarchean era, the Keivy domain was a median massif. Since the Keivy domain refers to structures that are often confined to hydrocarbon deposits in the Phanerozoic eon, it is easier to reason the occurrence of "methane" graphite in the Keivy crystalline schists [46,47].

Furthermore, the Lapland Belt of the proto-island arc type formed as the Belomorian domain interacted with domains to its northeast. The Titovka-Kolmozero (Kolmozero-Voronya) suture zone occurred, when the Kola-Norwegian and Murmansk domains interacted. This zone has features of both island arc and rift formations. Metabasites show these features as they are more akin to volcanites of young arcs in the Lapland-Kolvitsa Belt, while there is a shift towards mid-ocean ridge (MOR) volcanites in the Titovka-Kolmozero suture zone.

This paper provides data on the geodynamic evolution of the Archean structural and compositional complexes in the Kola region. Compared to the information that the specific volcanism in the early Precambrian era could predetermine authentic compositions of products at other stages of magmatism, we may consider the metallogenic forecast based on geodynamic reconstructions as promising for further research.

**Author Contributions:** N.E.K. conceived the research, designed experiments and prepared the considerable part of the manuscript. N.O.S. provided the metallogenic analysis and wrote the part on the regional metallogeny jointly with N.E.K., E.V.M. elaborated methods of geodynamic reconstructions and applied them to the current research. E.V.M. prepared the part on mathematical processing of data. All authors discussed the manuscript.

**Funding:** The research leading to this paper was funded by the Geological Institute of the Kola Science Centre RAS, public contract No. 0231-2015-0007.

**Acknowledgments:** The authors express their deepest gratitude to T.S. Marchuk for her highly skilled and qualified processing of the geochemical material and editing of the current paper. We also appreciate unknown reviewers for constructive comments and corrections.

**Conflicts of Interest:** The authors declare no conflict of interests.

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
