# Peer review of "Geodynamic Evolution and Metallogeny of Archaean Structural and Compositional Complexes in the Northwestern Russian Arctic"

_minerals, doi:10.3390/min8120573_

Round 1
Reviewer 1 Report
I have seen this document twice and consider that the improvements of the new version are not significant. Moreover the recommendations on the previous version have not been taken into account. Thus, it cannot be considered in the present form for publication.
Author Response
Dear Editor(s),
earlier we submitted our article entitled “Geodynamic Evolution and Metallogeny of Archean Structural and Compositional Complexes in the Northwestern Russian Arctic” to the Special (topical) Issue of your magazine “Arctic Mineral Resources: Science and Technology”. It was rejected at the first stage, but both reviewers admitted that it could be re-submitted upon serious processing in terms both of contents and language.
We attempted our best and forwarded the revised manuscript version. As we supposed, we took into account comments of the both reviewers to the fullest possible degree. Though, after that, we received new comments from them with strong criticism. We addressed the text again and tried to provide further more detailed clarifications where required. Herewith we enclose the modified version of the manuscript.
At the same time, if we almost completely agreed with the criticism contained in the text of primary reviews, this time we have issues with some comments, and we would like to express our opinion on the matter.
We cannot completely accept the criticism of the respected first reviewer, who says that we have not amended the original text. The portion of the text that concerns the mathematical data processing was almost completely reworked and expanded with necessary references as we believed. We also think that the predominance of references to Russian publications in the mathematical part cannot seriously hamper perception of the information since application of formulas in publications of any country is easily understandable for a wide range of experts. Evgeny Martynov, one of the coauthors, being a professional mathematician, many times reported results of his researches at conferences and meetings with International participation, and his findings were completely comprehended by the listeners regardless of the language.
We understand that this cannot be regarded as an argument for the respected reviewer and do not attempt to influence the evaluation of his work. We would only like to minutely explain our attempts in dealing with the comments since we cannot completely understand what the respected reviewer wanted to see in the mathematical part of the paper to recommend it for publishing.
We would like to mention here that such studies both in Russia and worldwide are not much common, and the approach to the data treatment is not a conventional one, which as we suppose created major difficulties in understanding of the material. This turned out complicated to find available references in International publications. We believe that the publication of this paper in your journal could become the first step in dismantling the existing language barrier. With regard to the comment on providing a detailed description of the applied method, including mathematical algorithms, we would like to stress that all the main methods and algorithms applied in this paper have been published before (also in International sources), and the corresponding references were added to the revised paper.
Here, we would like to enlarge upon the above points. This paper is not intended to describe the nuances of our method. It rather aims at throwing light on the ways to solve specific geological tasks using approaches that we propose. Hence, we revised it again, expanded the Methods section with clarification for easier assimilation of our unconventional principles and algorithms by an expert in any field of research.
Regarding the comments on geochronology, we also cannot agree in full. As the respected reviewer, we also know that various absolute ages exist for the same rock units. Moreover, we emphasize this in the introduction. Exactly this is what we use to justify the need to continue studying the matter and propose for this purpose the method, which is being mainly discussed with the reviewer. And we see no contradictions with his/her point of view. It is another matter that, to search for a generalized indicator (shown in Figure 3), we constructed some age sequences for the geological units, but on the basis of purely geological, not geochronological observations, references to which are given, and which have no set-based interpretations. If we, for example, would have data that the metavolcanites of the Lapland granulite belt formed as a result of interaction of two blocks, regardless of the age estimates, it is clear that they were younger than the rocks of the blocks, etc. The only exception is in the statement that the oldest rocks in Canada and Karelia are older than the rocks of the Kola region. We provided only one reference to prove it, but this fact is known to any researcher, who studied these regions, and this will be understood unambiguously. Nevertheless, we added another qualifying reference and provided some clarification in the paper on the matter. We would like to also mention that the paper contains references to Russian geochronological publications, but this cannot make the understanding difficult for a reader since the main information is contained in figures with isochrons comprehensible for experts from any country.
The comments of the second reviewer are clear to us, and we addressed them more specifically. With regard to the first comment on the methods, we refer to the explanations provided above. The other comments are considered in full.

Reviewer 2 Report
Please see the comments in the attached document!

Author Response
Dear Editor(s),
earlier we submitted our article entitled “Geodynamic Evolution and Metallogeny of Archean Structural and Compositional Complexes in the Northwestern Russian Arctic” to the Special (topical) Issue of your magazine “Arctic Mineral Resources: Science and Technology”. It was rejected at the first stage, but both reviewers admitted that it could be re-submitted upon serious processing in terms both of contents and language.
We attempted our best and forwarded the revised manuscript version. As we supposed, we took into account comments of the both reviewers to the fullest possible degree. Though, after that, we received new comments from them with strong criticism. We addressed the text again and tried to provide further more detailed clarifications where required. Herewith we enclose the modified version of the manuscript.
At the same time, if we almost completely agreed with the criticism contained in the text of primary reviews, this time we have issues with some comments, and we would like to express our opinion on the matter.
We cannot completely accept the criticism of the respected first reviewer, who says that we have not amended the original text. The portion of the text that concerns the mathematical data processing was almost completely reworked and expanded with necessary references as we believed. We also think that the predominance of references to Russian publications in the mathematical part cannot seriously hamper perception of the information since application of formulas in publications of any country is easily understandable for a wide range of experts. Evgeny Martynov, one of the coauthors, being a professional mathematician, many times reported results of his researches at conferences and meetings with International participation, and his findings were completely comprehended by the listeners regardless of the language.
We understand that this cannot be regarded as an argument for the respected reviewer and do not attempt to influence the evaluation of his work. We would only like to minutely explain our attempts in dealing with the comments since we cannot completely understand what the respected reviewer wanted to see in the mathematical part of the paper to recommend it for publishing.
We would like to mention here that such studies both in Russia and worldwide are not much common, and the approach to the data treatment is not a conventional one, which as we suppose created major difficulties in understanding of the material. This turned out complicated to find available references in International publications. We believe that the publication of this paper in your journal could become the first step in dismantling the existing language barrier. With regard to the comment on providing a detailed description of the applied method, including mathematical algorithms, we would like to stress that all the main methods and algorithms applied in this paper have been published before (also in International sources), and the corresponding references were added to the revised paper.
Here, we would like to enlarge upon the above points. This paper is not intended to describe the nuances of our method. It rather aims at throwing light on the ways to solve specific geological tasks using approaches that we propose. Hence, we revised it again, expanded the Methods section with clarification for easier assimilation of our unconventional principles and algorithms by an expert in any field of research.
Regarding the comments on geochronology, we also cannot agree in full. As the respected reviewer, we also know that various absolute ages exist for the same rock units. Moreover, we emphasize this in the introduction. Exactly this is what we use to justify the need to continue studying the matter and propose for this purpose the method, which is being mainly discussed with the reviewer. And we see no contradictions with his/her point of view. It is another matter that, to search for a generalized indicator (shown in Figure 3), we constructed some age sequences for the geological units, but on the basis of purely geological, not geochronological observations, references to which are given, and which have no set-based interpretations. If we, for example, would have data that the metavolcanites of the Lapland granulite belt formed as a result of interaction of two blocks, regardless of the age estimates, it is clear that they were younger than the rocks of the blocks, etc. The only exception is in the statement that the oldest rocks in Canada and Karelia are older than the rocks of the Kola region. We provided only one reference to prove it, but this fact is known to any researcher, who studied these regions, and this will be understood unambiguously. Nevertheless, we added another qualifying reference and provided some clarification in the paper on the matter. We would like to also mention that the paper contains references to Russian geochronological publications, but this cannot make the understanding difficult for a reader since the main information is contained in figures with isochrons comprehensible for experts from any country.
The comments of the second reviewer are clear to us, and we addressed them more specifically. With regard to the first comment on the methods, we refer to the explanations provided above. The other comments are considered in full.
Answers to the comments of Review-2:
1. Page 2, line 49: please give a reference concerning the application of petrochemical methods. Cleared, please see the paper.
2. Page 2, line 51: which quantitative indicators do you mean? Cleared, please see the paper.
3. Page 2, line 57: provided by… We believe that in this case the sentence has no predicate, which is grammatically incorrect.
4. Page 2, line 80: say formation instead of generation. We believe that these two words have equal meaning in this context.
5. Page 2, line 89: always use amphibolite-facies instead of amphibolite facies. Same for all other facies. We use a common form of spelling as the native speaker, who proofread the text and writes this word as an attributive chain with no hyphen.
6. Page 2, line 94: what do you mean by physical fields? Cleared, please see the paper.
7. Page 4, Figure 2: what are the variables (one of them presumably time) of this figure? Please refer to the paper. This figure illustrates the partial order notion. That is why coordinate axes convey no specific geological meaning. The only thing, which is relevant here, is that the space is two-dimensional. This means that it is enough to specify two arbitrary axes for introducing a coordinate system inside it. Thus, axis F describes the arrangement of rock assemblages towards each other with time.
8. Page 5, line 194: please never refer to something such as truth in a scientific paper… use high probability or something similar. We appreciate this comment. Cleared, please see the paper.
9. Page 6, line 204: how can you deduce based on figure 4 that the earth`s evolution was non-linear? We added the following clarification: The nonlinearity of the evolutional processes is attested by the following fact established in this research. As a result of studying the mutual arrangement of reference images (Phanerozoic basalts) and rock assemblages of Precambrian metabasalts in the feature space, it turned out that the shifting of their petrochemical features cannot be statistically reliably described using one linear function. This means that at least two linear functions have to be applied to describe such a shift as shown in Figure 4.
10. Page 6, line 209: well, I think as soon plate tectonics started you can use the actuality principle. Cleared, necessary adjustments are made in the text.
11. Page 6, line 219: what do you mean by silicate analyses? Do you mean bulk rock compositions instead? This study was based on more than 1100 bulk rock chemical analyses of Precambrian metamorphites and about 1100 bulk rock chemical analyses of Phanerozoic magmatic rocks, with which the comparison was made.
12. Page 7, Figure 5: what are the lables? What is the variable p you depict in this figure?
The caption to figure 5 was expanded with the following explanation: rФ for the proximity of a certain Phanerozoic rock assemblage represented by a set of figurative points in a space of chemical rock composition parameters to the simulated quadratic surface in relation to a chosen metrics, and rД for the proximity of a certain Precambrian rock assemblage to the surface respectively.
13. Page 7, line 264-265: what are Phanerozoic standards? We provided necessary explanations in the text: To identify the geodynamic setting, in which the studied Precambrian rock assemblage formed, we manipulated with combinations of figurative points for Precambrian rock assemblages, which were compared with the Phanerozoic standards located on the simulated interface. The groups of rocks formed from the Phanerozoic assemblages were considered as the references. These groups included structures evolved in various geodynamic settings. Juvenile arcs are represented by complexes developed over the oceanic-type crust, which are the South Sandwich Islands arc, Mariana island arc, and Tonga-Kermadec island-arc segment. The developed arcs of the transitional-type crust complexes are the Kuriles, the Kamchatka, and Aleut island arcs. The mature arcs are represented by the complexes developed over the continental-type crust being the Japanese, Lipari, and New Guinea island arcs. The island arcs are described by 557 chemical analyses of basalts. The mid-oceanic ridges (oceanic-type crust) are represented by the rocks of the Atlantic, Indian, and Pacific oceans and defined by 195 chemical analyses of basalts. The continental rifts are represented by the rocks of the Rhine graben, Baikal rift zone, West-American rift zone, African-Arabian rift belt, rifts of the East Asia, and Central-France massif. These structures are determined by 214 chemical analyses of basalts. The traps are represented by basalts and dolerites of the African and Indian platforms and studied in 176 chemical analyses.
14. Page 8, line 314: again you mean bulk rock compositions! Please see above.
15. Page 9, Table 1: please give at least one example in the text of how you derive a parameter such TRAP… Table 1 demonstrates sample proximity factors for the Precambrian rock assemblages towards the corresponding Phanerozoic standards, and namely TRAP, СК, MA, YA, DA, and MOR. The Euclidian distance between the vectors of the sample medians was used as a proximity factor for the Precambrian rock assemblages and Phanerozoic standards in the space of parameters for the chemical rock composition.
16. Page 11, Figure 8: please give names to the variables! Axis x: varying ratios of metavolcanites and metasedimentary rocks distribution, axis y: studied complexes where number of groups are numbered as in Figure 3. Constructed in two graphs, for the Murmansk Domain (group 5) and the Belomorian Mobile Belt (group 6). The names to the axes were given in the caption to Figure 8 and earlier in the text.
17. Page 11, line 370: what is continental-crustal matter??? Cleared, please see the paper.
18. Page 13, line 442: what do you mean by … suits, do you mean sequences? We believe that these words have similar meaning in this context. Cleared, please see the paper.
Please, go through the explanations to the comments provided above. We do cherish hopes for your positive decision.
The language has been carefully reworked with the help of the MDPI English Editing Service.
Yours sincerely,
Prof. Nikolay Kozlov and coauthors
16/11/18

Round 2
Reviewer 1 Report
I consider this article suitable for publication in Minerals.
Reviewer 2 Report
The revisons are fine!